# CHIR99021 Maintenance of the Cell Stemness by Regulating Cellular Iron Metabolism

**DOI:** 10.3390/antiox12020377

**Published:** 2023-02-04

**Authors:** Yingying Han, Yong He, Xiaofang Jin, Jiayi Xie, Peng Yu, Guofen Gao, Shiyang Chang, Jianhua Zhang, Yan-Zhong Chang

**Affiliations:** 1Laboratory of Molecular Iron Metabolism, Key Laboratory of Animal Physiology, Biochemistry and Molecular Biology of Hebei Province, Ministry of Education Key Laboratory of Molecular and Cellular Biology, College of Life Science, Hebei Normal University, Shijiazhuang 050024, China; 2College of Basic Medical Sciences, Hebei Medical University, Shijiazhuang 050017, China; 3Department of Automatic, Tsinghua University, Beijing 100084, China

**Keywords:** CHIR99021, glycogen synthesis kinase 3, classical Wnt signaling pathway, ferritin, Neuro-2a, cell stemness

## Abstract

CHIR99021 is an aminopyrimidine derivative, which can efficiently inhibit the activity of glycogen synthesis kinase 3α (GSK-3α) and GSK-3β. As an essential component of stem cell culture medium, it plays an important role in maintaining cell stemness. However, the mechanism of its role is not fully understood. In the present study, we first found that removal of CHIR99021 from embryonic stem cell culture medium reduced iron storage in mouse embryonic stem cells (mESCs). CHIR99021-treated Neuro-2a cells led to an upregulation of ferritin expression and an increase in intracellular iron levels, along with GSK3β inhibition and Wnt/GSK-3β/β-catenin pathway activation. In addition, iron treatment activated the classical Wnt pathway by affecting the expression of β-catenin in the Neuro-2a cells. Our data link the role of iron in the maintenance of cell stemness via the Wnt/GSK-3β/β-catenin signaling pathway, and identify intermediate molecules, including Steap1, Bola2, and Kdm6bos, which may mediate the upregulation of ferritin expression by CHIR99021. These findings reveal novel mechanisms of the maintenance of cell stemness and differentiation and provide a theoretical basis for the development of new strategies in stem cell treatment in disease.

## 1. Introduction

Embryonic stem cells (ESCs), derived from early embryos or primary gonads, have the abilities of self-renewal and pluripotency [1,2]. Oct4, Sox2, and Nanog jointly regulate the pluripotency of ESCs [3]. The Wnt signaling pathway is a very important pathway to affect the stem cell behavior, which contains a complex protein interaction network [4]. When the Wnt signal is absent, the scaffold proteins, Axin, adenomatous polyposis coli (APC), GSK-3β, and casein kinase 1α (CK1α), form a protein destruction complex, which induces β-catenin phosphorylation and then degradation by the proteasome. Conversely, the Wnt ligands bind to the Frizzled receptor and recruit the Dishevelled (DVL) protein, resulting in the dissociation of the destruction complex. Subsequently, β-catenin binds to T-cell factor/lymphoid enhancer factor (TCF/LEF) to activate the target genes [5]. The non-canonical Wnt pathway does not require β-catenin. This alternate signaling cascade activates multiple intracellular targets. The most extensively studied are the Wnt/planar cell polarity (PCP) and Wnt/calcium (Ca^2+^) pathways [6,7,8].

The canonical Wnt/β-catenin signaling pathway plays a key role in embryonic development and stem cell stemness. The activation of this pathway has been shown to increase the expression of pluripotent genes in mESCs. Besides, the activation of the pathway can significantly up-regulate the expression of prostate tumor stem cell markers B-cell-specific Moloney murine leukemia virus integration site 1 (BMI-1), Aldehyde dehydrogenase 1A1 (ALDH1A1), Cluster of differentiation-44 (CD44), aldehyde dehydrogenase 1 family, member A3 (ALDH1A3), and SOX2 mRNA [9]. Additionally, studies in colorectal cancer have found that transmembrane 4L6 family member 1 (TM4SF1) can maintain cancer cell stemness and epithelial-mesenchymal transformation (EMT) through the Wnt/β-catenin/c-Myc/SOX2 pathway [10].

Iron is an element essential to normal physiological functions and metabolic activities [11]. Most of the iron is liganded within functional proteins, including hemoproteins, iron-sulfur (Fe/S) proteins, and enzymes which contain non-heme, non-Fe/S iron. In addition, there is a fraction of uncommitted iron, called the labile iron pool (LIP). These different forms of iron can participate in numerous of important biological processes, such as oxygen transport, electron transport, gene expression, cell proliferation, and cell differentiation [12]. Both iron deficiency and excess can have harmful effects. Therefore, it is essential to maintain iron homeostasis to support the normal physiological activities, while protecting against the toxicity of the catalytic metal. Iron levels are tightly controlled by some proteins that regulate the absorption, storage, circulation, and utilization of iron [12,13]. Cellular iron uptake is mainly mediated by the plasma iron transport protein, transferrin (Tf), and its receptor, transferrin receptor 1 (TfR1). Tf-bound iron enters to the LIP, with some of this iron binding to cytosolic poly (rC) binding proteins PCBP1 and PCBP2, which can transport the metal to the ubiquitous iron storage protein, ferritin, or other non-heme iron-proteins. Excess iron can be transported out of cells by ferroportin 1 (FPN1) [14]. Cellular iron homeostasis is regulated by a post-transcriptional regulation mechanism mediated by iron regulatory proteins (IRPs). These two RNA-binding proteins specifically bind to the conserved motif, iron response element (IRE), in the untranslated regions (UTRs) of mRNAs encoding iron metabolism-related proteins [15].

The fate of stem cells is regulated by many factors [16], among which the endogenous factors mainly include the expression of genes, while the exogenous factors mainly include differentiation and inhibition between cells and the effects of external substances. As an exogenous factor, iron also affect the stemness and differentiation of stem cells. Metal ions, such as Mn, Co, Al and Fe, can regulate cell attachment and affect neuronal differentiation [17,18]. Besides, it was found that the neural differentiation of mESCs decreased upon addition of different concentrations of iron oxide nanoparticles to retinoic acid-induced mESCs [19]. In human dental pulp stem cells, the iron chelator, deferoxamine (DFO), can promote the expression of proteinaceous factors related to the stem cell characteristics [20,21,22]. Besides, at both cellular level and in clinical treatment, iron also play vital roles in the normal function of mesenchymal stem cells [23].

Both iron and the Wnt/GSK-3β/β-catenin pathway play a role in the maintenance of cell stemness. In fact, the iron and GSK-3β pathways may intersect. It has been found that Deferasirox (DFX) can trigger the proteolysis of cyclin D1 in mantle cell lymphoma (MCL), which requires the participation of GSK-3β [24]. In a human neuroblastoma cell line (SH-SY5Y), FeSO_4_ can activate GSK-3β, and down-regulate the phosphorylation of GSK-3β [25]. In addition, in APP/PS1 transgenic mice, intranasal administration of DFO can abrogate the activity of GSK-3β, thus inhibiting tau protein phosphorylation [26]. In Parkinson’s disease dementia (PDD), the neurotoxicity resulted by iron can promote the activation of GSK-3β-related pathways, thus playing an important role in the pathological synergism of α-syn, tau and Aβ [27]. These studies suggest that there may be a specific relationship between iron and GSK-3β, however it is unclear how GSK3 may affect iron homeostasis. In this study, we investigated the effects of CHIR99021 on cellular iron homeostasis. Our data demonstrate that iron has an effect on cell stemness and that this effect is associated with the Wnt/GSK-3β/β-catenin pathway. Our findings provide insight into hitherto unknown regulatory mechanisms controlling stem cell stemness maintenance and differentiation.

## 2. Materials and Methods

### 2.1. Cell Culture

The Neuro-2a cell line, a mouse neuroblastoma cells, was cultured in DMEM (GIBCO, Grand Island, New York, USA 8121382) with 10% fetal bovine serum (Biological Industries, Cromwell, CT, USA) and 1× Pen/Strep (GIBCO, 10378016). Cells were maintained at 37 °C in a humidified incubator containing 5% CO_2_ (Thermo Fisher Scientific, Waltham, MA, USA).

### 2.2. Preparation of MEF Feeder Cells and Culture of mESCs

Healthy ICR male and female mice aged 2–3 months were selected to mate, and primary mouse embryonic fibroblast (MEF) feeder cells were obtained from pregnant mice at day 12.5 or 13.5 postfertilization. The embryonic day 12.5 (E12.5) or E13.5 fetal mice were isolated, and the head, limbs, tail, and internal organs were removed, leaving only the torso. The tissue fragments were then cut to <1 mm^3^ pieces and digested into individual cells with 0.25% trypsin. Next, the cells were cultured in MEF feeder culture medium, which contained DMEM supplemented with 10% FBS, 1× nonessential amino acids (GIBCO, 11140), 1× L-glutamine (Specialty media, TMS-002-C) and 1× Pen/Strep. MEF feeder cells of generation 2–3 were incubated with MEF feeder culture medium with mitomycin C (10 μg/mL) for 2–3 h to inhibit proliferation. Afterwards, the cells were cryopreserved until subsequent use in embryonic stem cell culture experiments.

The mESCs were generated and banked in our laboratory. A 0.1% gelatin was added to cover the wells of 24-well plates and incubated in a tissue culture incubator for 2–3 h in advance of use. The MEF feeder cells were then resuscitated and seeded onto the gelatin-coated plates until the cells covered the bottom. Additionally, mESCs were then resuscitated and maintained in KnockOut^TM^ DMEM (GIBCO, 10829018) supplemented with 15% KnockOut^TM^ serum replacement (GIBCO, A3181502), 1× nucleosides for ES cells (Merck Millipore, ES-008-D), 1× nonessential amino acids, 1× L-glutamine, 1× 2-mercaptoethanol (GIBCO, 21985023), 1× Pen/Strep, 1000 U/mL LIF (GIBCO, A35933), 3 μM CHIR99021 (Selleck, Houston, TX, USA S2924), and 1 μM PD0325901 (Selleck, S1036).

### 2.3. Drug Treatment

Cells were treated with CHIR99021, deferoxamine (DFO, Sigma, D9533) or ferric ammonium citrate (FAC, Sigma, St. Louis, MO, USA F5879). The concentrations and treatment times were as indicated in the results and figures.

### 2.4. Western Blot Analysis

Cells of the control group and drug-treated group were lysed in RIPA lysis buffer containing 95% RIPA buffer (Solarbio, Beijing, China R0010), 2% protease inhibitors (Roche, 04693116001), 2% phosphatase inhibitors (Roche, Basel, Switzerland, 04906837001), and 1% PMSF (Solarbio, P0100) for 30 min, shaken every 5 min. The protein concentration was assessed by a BCA kit (Yeasen, Shanghai, China, 20201ES76). Protein samples were resolved by SDS-PAGE (10% or 12% acrylamide) at 25 μg per lane and then transferred onto nitrocellulose (NC) membranes, which were cut appropriately, and blocked in 5% skimmed milk for 1.5 h and then incubated with different primary antibodies overnight at 4 °C. The next day, after washing, the membranes were incubated with the corresponding secondary antibodies at room temperature for 1.5 h. Finally, we used an ECL kit (CWBIO, CW0049M) and a chemiluminescence imager (Bio-Rad, Hercules, CA, USA) for visualization. Quantitative analysis of the protein bands was normalized to β-actin, as an internal reference.

The primary and corresponding secondary antibodies we used were as follows: mouse anti-TfR1 (1:5000, 13-6890, Invitrogen, Waltham, MA, USA), rabbit anti-FPN1 (1:8000, MTP11-S, Alpha Diagnostic International, San Antonio, TX, USA), rabbit anti-ferritin heavy chain (1:5000, ab183781, Abcam, Cambridge, UK), rabbit anti-ferritin light chain (1:5000, ab109373, Abcam), rabbit anti-DMT1(+IRE) (1:5000, NRAMP21-S, Alpha Diagnostic International, USA), rabbit anti-DMT1(-IRE) (1:5000, NRAMP23-S, Alpha Diagnostic International, USA), mouse anti-GSK-3α/β (1:5000, sc-7291, Santa Cruz, Dallas, TX, USA), rabbit anti-p-GSK3β (Ser9) (1:5000, #9336, CST, Danver, MA, USA), rabbit anti-β-catenin (1:2000, 610153, BD), rabbit anti-NCOA4 (1:5000, Abbkine, Wuhan, China), rabbit anti-cyclin D1 (1:2000, #2922, CST, USA), rabbit anti-Ki67 (1:10,000, ab15580, Abcam), mouse anti-β-actin (1:10,000, CW0096, CWBIO, Beijing, China); goat anti-mouse lgG (H+L) (1:10,000, RS0001, Immunoway, Plano, TX, USA), and goat anti-rabbit lgG (H+L) (1:10,000, RS0002, Immunoway, USA).

### 2.5. Total RNA Extraction and qRT-PCR

Total RNA was extracted from neuro-2a cells or mESCs using TRIzol reagent (Invitrogen, 15596018). RNA was then reverse-transcribed to cDNA. Next, we mixed 10 μL SYBR Green PCR Master Mix (CWBIO, CW0957), 1μg cDNA, 0.4 μL forward primer, 0.4 μL reverse primer and 5.2 μL RNase-free water for the subsequent PCR reaction. qRT-PCR amplification was performed using a Bio-RAD CFX96 Connect Real-Time System. The primer sequences used are listed in Table 1.

### 2.6. RNA-Sequencing (RNA-Seq) and Data Analysis

The Neuro-2a control group cells and CHIR99021-treatment group samples were collected in TRIzol (Invitrogen, 15596018) and sent to Majorbio (Shanghai, China). The original data and results of RNA-sequencing were analyzed by Majorbio.

### 2.7. Inductively Coupled Plasma Mass Spectrometry (ICP-MS)

Total cellular iron content was evaluated by ICP-MS. Cell samples were centrifuged and collected after 0.25% trypsin digestion. Then, the samples were digested in 500 μL 65% nitric acid at room temperature overnight. The nitric acid was evaporated by heating for 20 min in a metal bath at 90 °C. Afterwards, 500 μL 30% H_2_O_2_ was added and allowed to react at 70 °C for 15 min, and then continued to vaporize for at least 6 h at 100 °C until the liquid in the tube was almost fully volatilized. The digested samples were re-suspended and mixed in 1 mL ultra-pure water for subsequent ICP-MS.

### 2.8. Alkaline Phosphatase Staining

For mESCs, the BCIP/NBT Alkaline Phosphatase Color Development kit (Beyotime, Jiangsu, China, C3206) was used according to the manufacturer’s instructions. Cells were carefully washed 3–5 times with PBS and fixed with an appropriate amount of 4% formaldehyde for 20 min. Next, cells were incubated with the BCIP/NBT working solution for 15 min in the dark. An appropriate amount of double-distilled water was added to the plates to wash the cells 1–2 times to stop the color reaction. The results were examined and imaged under a light microscope (OLYMPUS, FV300).

### 2.9. Electric Cell-Substrate Impedance Sensing (ECIS) Detection of Cell Proliferation

The proliferation of cells was detected using an ECIS instrument (Applied BioPhysics, Troy, NY, USA). The experimental materials and reagents prepared include: 0.22 μm filter, syringe, sterile water, and 100 mM cysteine. An 8W10E+ electrode plate was pre-treated with 400 μL cysteine solution at a final concentration of 10 mM in each well, and incubated at 37 °C overnight. The electrode plate was washed 3 times with sterile water the next day. The cells were then seeded onto the electrode plate with about 1.5 × 10^4^ cells per well, and left for 5 min. The real-time detection was performed after cells placed in the 37 °C incubator and connected to the ECIS instrument.

### 2.10. Profiles of Mitochondrial Respiration and Glycolysis

An Oxygraph-2k (O2k, TissueGnostics Asia Pacific Limited, Beijing, China) was used to assess the degree of oxidative phosphorylation and glycolysis of cell samples. The cells were digested, centrifuged, resuspended and counted, and then added into the chambers of the instrument. For the levels of oxidative phosphorylation, oxygen consumption rate (OCR) was recorded after injections of pyruvate (P), malic acid (M), ADP (D), oligomycin (Omy), the mitochondrial uncoupling agent, FCCP (U+), and antimycin A (Ama). For the level of glycolysis, the extracellular acidification rate (ECAR) was recorded, after injections of glucose (Glu), oligomycin (Omy), and 2-deoxy-glucose (2-DG), according to the manufacturer’s instructions. The data were exported and analyzed by DatLab (New York, NY, USA, 7.4.0.4) software.

### 2.11. Statistical Analysis

All data are presented as the mean ± standard error (SEM). The statistical graphs were generated using GraphPad Prism 6 software. Differences between two groups were compared by non-paired *t*-tests. Differences were considered statistically significant when *p* < 0.05.

## 3. Results

### 3.1. The Effects of CHIR99021 and Iron on Cell Stemness in mESCs

The small molecule compound, CHIR99021, is an aminopyrimidine derivative which can efficiently inhibit GSK-3α and GSK-3β. It is also commonly used to culture ESCs and induced pluripotent stem cells (iPSCs). To explore the effects of CHIR99021 on iron in mESCs, we used qRT-PCR to examine the transcriptional levels of H-ferritin (FtH) and L-ferritin (FtL), which make up the ubiquitous intracellular iron storage protein. The expression levels of *FtH* mRNA were significantly decreased after the removal of CHIR99021 alone or with simultaneous removal of LIF, PD0325901, and CHIR99021 (Figure 1A). Likewise, the expression levels of *FtL* mRNA were also significantly diminished after removal of CHIR99021 alone (Figure 1B). There was no significant difference in the expression levels of *Oct4* mRNA after removing CHIR99021 alone or removing LIF, PD0325901 and CHIR99021 (Appendix A), while the levels of *Sox2* mRNA were significantly decreased (Appendix A). These results demonstrated that removal of CHIR99021 reduced iron storage in mESCs.

To further examine the relationship between CHIR99021, iron, and cell stemness in mESCs, we supplemented normal embryonic stem cell medium and embryonic stem cell differentiation medium, in which LIF, PD0325901, and CHIR99021 were simultaneously removed, with 25 µM ferric ammonium citrate (FAC) and examined the mRNA levels of the pluripotency genes, *Oct4* and *Sox2*. Supplementing the embryonic stem cell culture medium with iron did not affect the expression of these mRNAs. However, the levels of *Sox2* mRNA were significantly increased after supplementing the differentiation medium with 25 µM FAC (Figure 1C,D). By alkaline phosphatase staining, we also found that the stemness was weakened in mESCs cultured with differentiation medium, with the cells exhibiting a differentiated, spreading morphology. After supplementing the differentiation medium with 25 μM, 50 μM, or 100 μM FAC, although the mESCs also appeared to differentiate, the color of the cells was darker than that of the DV- group (Figure 1E,F). Together, these results suggested that removal of CHIR99021 decreases iron levels and the stemness of mESCs; supplementation with iron may be beneficial to the maintenance of stemness.

### 3.2. CHIR99021 Increasing the Expression of Ferritins, Cellular Iron Levels and Inhibiting the Expression of FPN1 in Neuro-2a Cells

In order to further explore the effects of CHIR99021 on iron metabolism, we treated Neuro-2a cells with CHIR99021 and assessed the expression of iron metabolism related proteins, including FtH, FtL, TfR1, FPN1, and DMT1 (+/-IRE) (divalent metal transporter 1). The protein levels of FtH and FtL increased significantly in CHIR99021-treated Neuro-2a cells (Figure 2A,B). The levels of *FtH* and *FtL* mRNA were consistent with the increased protein levels (Figure 2C). As ferritin is the main intracellular iron storage protein, we next investigated whether total intracellular iron levels were changed by ICP-MS; total intracellular iron levels increased significantly after CHIR99021 treatment (Figure 2D). We also evaluated the levels of TfR1 protein and mRNA and found no significant differences (Figure 2E–G). However, the protein and mRNA levels of FPN1 [28] decreased significantly (Figure 2E,F,H). Additionally, the protein level of DMT1 decreased significantly (Appendix A). These results indicated that iron metabolism has been affected, and iron export decreased, which is expected to increase the amount of uncommitted iron in cells and, in turn, stimulate an increase in ferritin expression.

### 3.3. DFO Abrogates the Increase of Ferritin Induced by CHIR99021

In order to further confirm that CHIR99021 increases cellular iron levels, we co-treated cells with CHIR99021 and the iron chelator, DFO. We initially performed a concentration-dependence experiment, using 0–40 µM DFO. We found that the protein levels of FtH and FtL were decreased in the presence of 5 µM DFO (Figure 3A–D). Therefore, we proceeded to treat cells with 5 µM DFO and 3 µM CHIR99021. As previously shown, CHIR99021 treatment significantly increased the levels of FtH and FtL, while the expression of these proteins significantly decreased when DFO was included in the incubation (Figure 3E–H). These results suggested that the sequestration of iron could alleviate the increase of ferritin expression induced by CHIR99021.

### 3.4. CHIR99021 Stimulates the Expression of Steap1, Bola2 and Kdm6bos Increasing in Neuro-2a Cells

In order to further explore the effect of CHIR99021 treatment on iron homeostasis and the mechanism of increased ferritin expression, we performed RNA-seq analysis in cells treated with or without CHIR99021. As shown in Figure 4A, we identified 1364 differentially expressed genes between the two groups, of which 580 genes were up-regulated and 784 genes were down-regulated. Gene Ontology (GO) enrichment analysis categorizes groupings of genes into three general areas: biological process, cellular component, and molecular function. From the results of this type of analysis, we found most differentially expressed genes to be concentrated in cellular processes: biological regulations and metabolic processes of biological processes; the composition of cells and organelles in cellular components; and molecular binding parts in molecular functions (Figure 4B). The Kyoto Encyclopedia of Genes and Genomes (KEGG) database divides biological metabolic pathways into six areas, including metabolism, genetic information processing, environmental information processing, cellular processes, biological systems, and human diseases. We found that the differentially expressed genes were more heavily distributed in the process of translation and signal transduction (Figure 4C). These results helped us to hone in on the genes targeted by CHIR99021.

Next, based on the results of sequencing and alignment, we determined the genes which both |log2^FC^| ≥ 1 and were related to iron or metal ion binding from 1364 differentially expressed genes. Six-segment transmembrane epithelial antigen of prostate 1 (STEAP1) has oxidoreductase activity and metal ion binding function in the molecular functional grouping enriched by GO, and also plays a key role in iron homeostasis [31]. BolA-like protein (Bola2) participates in the assembly of iron-sulfur clusters on biological process grouping [32,33]. KDM6B is a KDM1 lysine-specific demethylase 6B, which has methyltransferase activity and metal ion binding function in the molecular function grouping. By qRT-PCR, we confirmed that the mRNA levels of *Steap1*, *Bola2* and *Kdm6bos* increased significantly upon CHIR99021 treatment (Figure 4D–F).

### 3.5. Both CHIR99021 and Iron Promote the Expression of β-Catenin

The small molecular compound, CHIR99021, is a highly effective inhibitor of GSK-3 α/β, while β-catenin is a downstream target gene of GSK-3β in the classical Wnt pathway [34]. To verify the effect of CHIR99021 on the Wnt/GSK-3β/β-catenin pathway, we examined the related proteins. We found that CHIR99021 significantly downregulated the expression of GSK-3α/β, concomitantly upregulating the levels of p-GSK3β and β-catenin (Figure 5A,B,D,E). Thus, the ratio of p-GSK3β/GSK-3β increased significantly with the treatment of CHIR99021 (Figure 5C). In addition, we found that 25 µM and 50 µM FAC significantly increased the expression of β-catenin (Figure 5F–K). In consideration of these and the previous results, we hypothesized that there may be a relationship between β-catenin and iron level, particularly on the context of CHIR99021 exposure.

### 3.6. CHIR99021 Affects the Growth of Neuro-2a Cells

Our above results demonstrate that CHIR99021 treatment altered intracellular iron homeostasis, including changes in iron metabolism-related molecules, as well as significant increases in intracellular iron levels. Among its many roles in enzymes and other proteins, iron participates in the synthesis and replication of DNA, thus playing important roles in the process of cell growth [35]. We therefore proceeded to examine the expression of cyclin D1 and Ki67 proteins. Treatment of cells with CHIR99021 led to significant decreases in both of these proteins (Figure 6A–C). In addition, we found by electric cell-substrate impedance sensing that CHIR99021 inhibits cell proliferation (Figure 6D).

### 3.7. CHIR99021 Treatment Decreases Mitochondrial Oxidative Phosphorylation and Enhances Glycolytic Capacity in Neuro-2a Cells

Oxidative phosphorylation (OXPHOS) and glycolysis are two major pathways of energy metabolism in mammalian cells. Normally, these two pathways are regulated to adapt to changes of external environment. As described above, we found that the GSK3 inhibitor, CHIR99021, can not only alter cellular iron levels and iron metabolism-related proteins, but also activate the Wnt/GSK-3β/β-catenin pathway. Both iron and this Wnt pathway could affect the maintenance of cell stemness [9,23,36]. When stem cells are in an undifferentiated state, mitochondria are immature and perinuclearly distributed, with lower OXPHOS capacity and higher glycolytic activity. Importantly, cellular energy metabolism patterns can also regulate cell fate through epigenetic mechanisms [37,38]. Therefore, we used the cellular energy metabolism analyzer, O2k, to evaluate cellular oxygen consumption and glycolytic capacity. The overall mitochondrial metabolic capacity decreased in cells treated with CHIR99021 (Figure 7A,B); basal respiration, ATP production-related oxygen consumption, and maximal respiration were all significantly decreased. Spare respiratory capacity, which represents the potential responsiveness of cells to energy demands, showed a downward trend (Figure 7C). In contrast, the overall glycolytic capacity of the cells was increased after CHIR99021 treatment (Figure 7D), while the maximum glycolytic capacity and glycolysis reserve capacity were all significantly increased (Figure 7E). These results suggest that CHIR99021 treatment diminishes cellular OXPHOS levels and enhances cellular glycolytic capacity, which is consistent with the characteristic that cells in a state of stemness mainly rely on glycolysis rather than OXPHOS.

## 4. Discussion

A multifunctional protein, GSK3, is involved in a variety of cellular processes, including proliferation, metabolism and embryonic development [39]. It contains GSK-3α and GSK-3β. The latter is one of the main members of the canonical Wnt pathway. Recent studies have revealed that this pathway plays an important role in the maintenance of stemness [9,10,40]. As an essential element, iron also has many biological functions. Among these, iron has been identified as a key factor in the maintenance of cancer stem cell (CSC) stemness. The iron chelation can inhibit the expression of the stem cell marker, Nanog, thus revealing a new treatment strategy for cancer [41]. However, it is not clear whether there is a relationship between GSK3 and iron, and, if so, what that mechanism may entail.

CHIR99021, a small molecule compound, can efficiently inhibit both GSK3. The compound is also commonly used in the culture of stem cells. In this study, we found that the expression of stemness-related genes and the transcription level of FtH and FtL in mESCs were significantly decreased upon removal of CHIR99021 from mouse embryonic stem cell culture medium. On the other hand, iron supplementation of embryonic stem cell differentiation medium was beneficial to the maintenance of mESCs stemness. In fact, we found that CHIR99021 can affect cell stemness by affecting intracellular iron levels. To further explore the relationship between GSK3 and cellular iron homeostasis, we treated Neuro-2a cells with CHIR99021, which led to increased expression of FtH and FtL at both the mRNA and protein levels, as well as increased intracellular iron content. Cellular iron homeostasis is regulated at the post transcriptional level by the IRE/IRP system. When intracellular iron levels increase, IRPs are unable to bind to the IREs in the 5′-UTR of the mRNAs encoding ferritin and FPN1, thus permitting their translation, while binding to the 3′-UTR in the mRNA encoding TfR1 is required to stabilize the message, so TfR1 levels decrease under elevated cellular iron conditions. Regulation occurs in the opposite direction upon depletion of cellular iron [42]. Our finding of increased ferritin, with concomitant decreases in FPN1 and unchanged TfR1 after treatment with CHIR99021, indicates that the changes in iron homeostasis induced by CHIR99021 may not be regulated by the system. However, treatment of cells with iron chelation and CHIR99021 alleviated the increase of ferritin expression induced by CHIR99021. Thus, we conclude that CHIR99021 indeed affects cellular iron homeostasis.

To further explore the mechanism of ferritin up-regulation by CHIR99021, we used RNA-seq and narrowed the differentially regulated genes based on iron or metal iron binding, finally identifying three genes of interest: *Steap1*, *Bola2* and *Kdm6bos*. The mRNA expression of all three of these genes increased significantly after CHIR99021 treatment. Interestingly, STEAP1 is highly expressed in prostate cancer [43]. In 2016, Kim et al. reported the presence of a single heme b prosthetic group in STEAP1, which can reduce metal ion complexes and oxygen. Diferric transferrin (Fe_2_Tf) binds to TfR1, which mediates iron uptake into cells. Then, Fe^3+^ is released from Tf and must be reduced prior to export from the endosome by the Fe^2+^ transporter, DMT1. This reduction can be mediated by STEAP1. After export from the endosome, iron can be incorporated into functional proteins or stored in ferritin [31]. Therefore, the increase of ferritin expression induced by CHIR99021 treatment may be a consequence of increased levels of STEAP1. Bola2 has been shown to bind to the multi-functional binding protein PCBP1 to form an intermediate iron chaperone complex and participate in the assembly of [2Fe-2S] clusters, while PCBP1 may bind iron in the LIP to transfer iron to ferritin to form a PCBP1-Fe-GSH-BolA2 complex [32,33]. Therefore, there may be a connection between ferritin and BolA2. KDM6B is a KDM1 lysine-specific demethylase with methyltransferase activity and metal ion binding function. KDM6B requires Fe^2+^ as a cofactor [44]. Together, our data provide support for a link between GSK3 activity and ferritin synthesis, possibly via the proteins encoded by the above genes, however, the specific mechanisms of these processes require further study.

As one of the main members of Wnt/β-catenin pathway, GSK-3β can phosphorylate β-catenin and cause it to be degraded by proteasome. When the phosphorylation of GSK-3β at Ser9 is increased, the activity of GSK-3β decreases [34]. Our results show that, with CHIR99021, the levels of GSK-3α/β were significantly decreased, while the p-GSK3β/GSK-3β ratio was significantly increased, ultimately leading to promotion of the expression of the downstream target molecule β-catenin. Thus, the Wnt/GSK-3β/β-catenin pathway was significantly activated. We therefore asked, since iron levels increased upon CHIR99021 treatment, could iron also affect the expression of β-catenin? To this end, we treated cells with FAC, which promoted the expression of β-catenin. The finding is consistent with a report demonstrating a positive correlation between FtL and β-catenin in glioma cells [45].

Iron can involve in the regulation of cell growth. Both iron deficiency and excess can affect the cell cycle, which may then affect a series of critical organismal processes [46,47]. Our data that cell proliferation was inhibited with the treatment of CHIR99021 is consistent with these properties of iron in mammals.

Iron has also been found to play a biological role in maintaining the stemness of mesenchymal stem cells and cancer stem cells [23,41]. The activation of the classical Wnt pathway has been reported to inhibit the differentiation of stem cells [9,10,36]. Additionally, in stem cells, the mode of energy metabolism mainly depends on glycolysis rather than oxidative phosphorylation. On the other hand, the mode of cell energy metabolism can also determine the fate of stem cells, with respect to stemness or differentiation, through epigenetic regulation [37,38]. We found that oxidative phosphorylation decreased, along with increased glycolysis, upon treatment with CHIR99021.

## 5. Conclusions

In summary, our study uncovers a relationship between GSK3 and ferritin, along with alterations in cellular iron metabolism that affect cells stemness and proliferation. These findings not only open new avenues for further exploration of stemness maintenance and differentiation, but also provide an insight for the treatment of stem cells in disease.

## Figures and Tables

**Figure 1 antioxidants-12-00377-f001:**
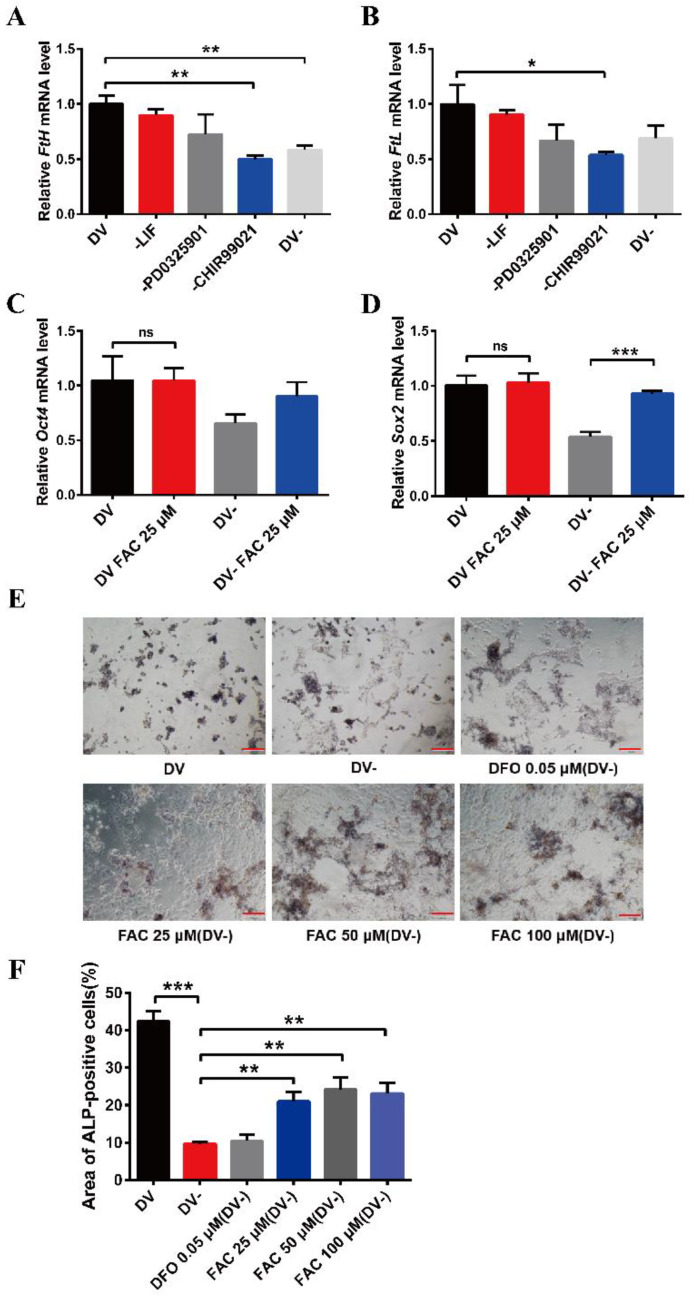
Effects of removal CHIR99021 and iron on the cell stemness in mESCs. (**A**,**B**) qRT-PCR analysis of *FtH* mRNA levels (**A**) and *FtL* mRNA levels (**B**) after incubation for 72 h in full derivation medium (DV) or without LIF, PD0325901, CHIR99021 or all three combined (DV-). (**C**,**D**) qRT-PCR analysis of pluripotency genes *Oct4* (**C**) and *Sox2* (**D**) expression after incubation for 72 h in full derivation medium or without LIF, PD0325901 and CHIR99021 (DV-), without or with the addition of 25 μM FAC. The mRNA levels are normalized to GAPDH mRNA levels. (**E**) Alkaline phosphatase (ALP) staining of mESCs treated for 72 h in full derivation medium (DV) or without LIF, PD0325901 and CHIR99021 (DV-), without or with the indicated amounts of FAC. Scale bar = 100 μm. (**F**) Quantification of area of ALP-positive cells (%). The data are expressed as the mean ± SEM. * *p* < 0.05, ** *p* < 0.01, *** *p* < 0.001, ns, not significant.

**Figure 2 antioxidants-12-00377-f002:**
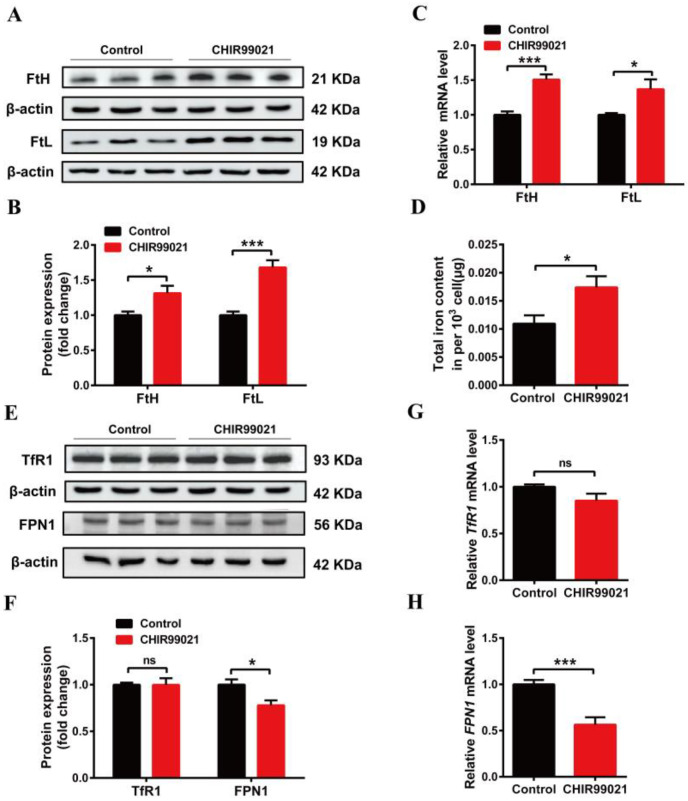
Changes in iron metabolism in Neuro-2a cells treated with CHIR99021. (**A**,**E**) Western blot analysis of FtH and FtL (**A**), TfR1 and FPN1 (**E**) protein after the cells were treated with or without 3μM CHIR99021 [29,30] for 72 h. (**B**,**F**) Quantification of Western blot data of FtH and FtL (**B**), TfR1 and FPN1 (**F**). (**C**,**G**,**H**) qRT-PCR analysis of *FtH* and *FtL* (**C**), *TfR1* (**G**), and *FPN1* (**H**) mRNA levels. (**D**) ICP-MS detection of total cellular iron levels in Neuro-2a cells treated with or without CHIR99021. The relative expression levels are normalized to β-actin and then expressed as the fold of control group. The data are presented as the mean ± SEM. * *p* < 0.05, *** *p* < 0.001, ns, not significant.

**Figure 3 antioxidants-12-00377-f003:**
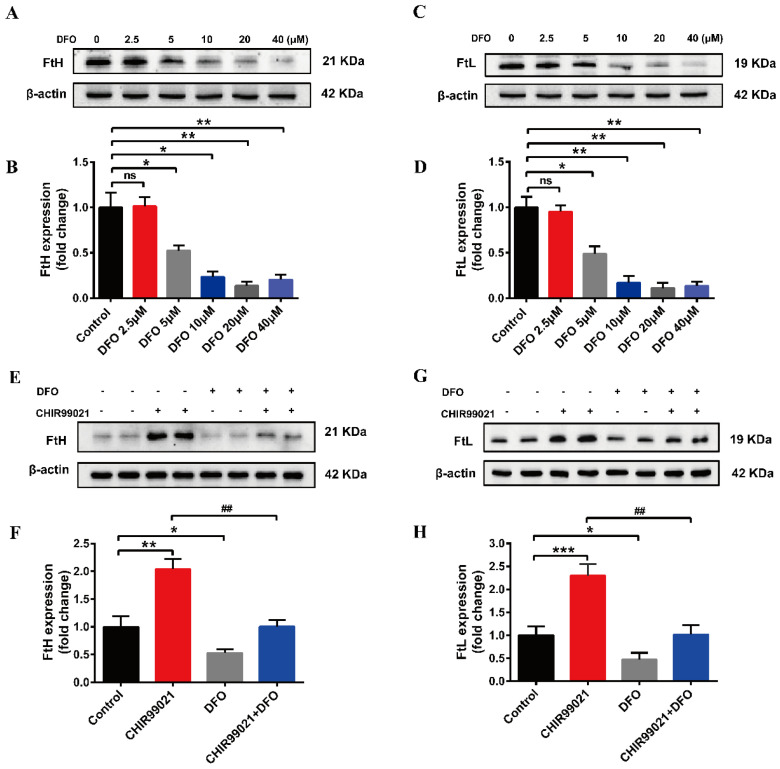
DFO prevents the increase of ferritin expression induced by CHIR99021. (**A**–**D**) Western blot analysis of FtH and FtL protein after treatment with the indicated concentrations of DFO. (**E**,**F**) Western blot analysis of FtH levels after treatment with or without CHIR99021 and/or DFO. (**G**,**H**) Western blot analysis of FtL levels after treatment with or without CHIR99021 and/or DFO. The relative expression levels are normalized to β-actin. The data are expressed as the mean ± SEM. * *p* < 0.05, ** *p* < 0.01, *** *p* < 0.001, ^##^
*p* < 0.01, ns, not significant.

**Figure 4 antioxidants-12-00377-f004:**
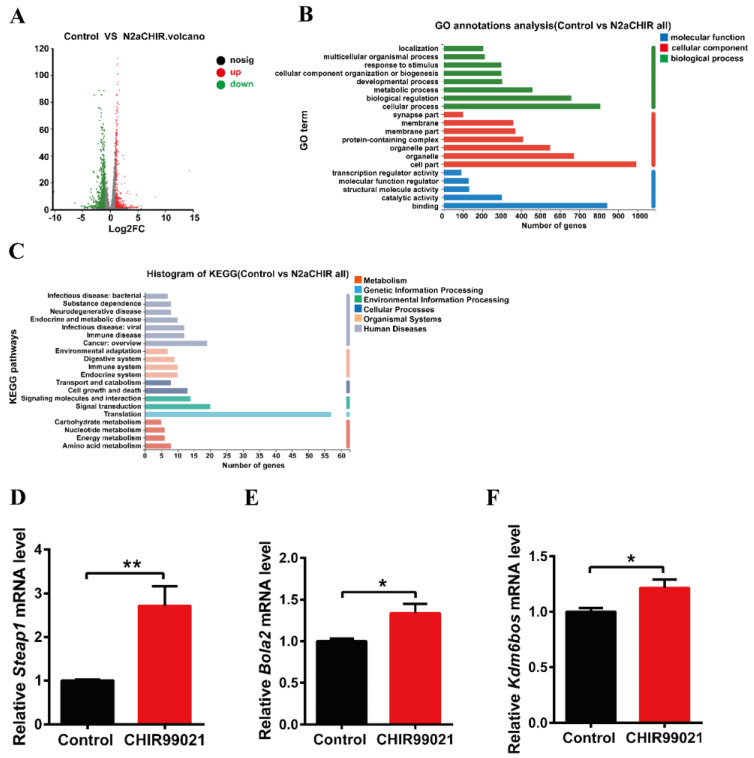
Comparison of RNA-seq results and verification of the differentially expressed genes between control and CHIR99021-treated cells. (**A**) Volcano plot of mRNA expression differences. (**B**) GO enrichment analysis between control and CHIR99021-treated Neuro-2a cells. (**C**) Statistical classification analysis of KEGG pathways between control and CHIR99021-treated Neuro-2a cells. (**D**–**F**) qRT-PCR analysis of *Steap1* (**D**), *Bola2* (**E**) and *Kdm6bos* (**F**) mRNA levels with or without CHIR99021 treatment. The mRNA levels are normalized to β-actin mRNA levels and then expressed as the fold of control group. The data are expressed as the mean ± SEM. * *p* < 0.05, ** *p* < 0.01.

**Figure 5 antioxidants-12-00377-f005:**
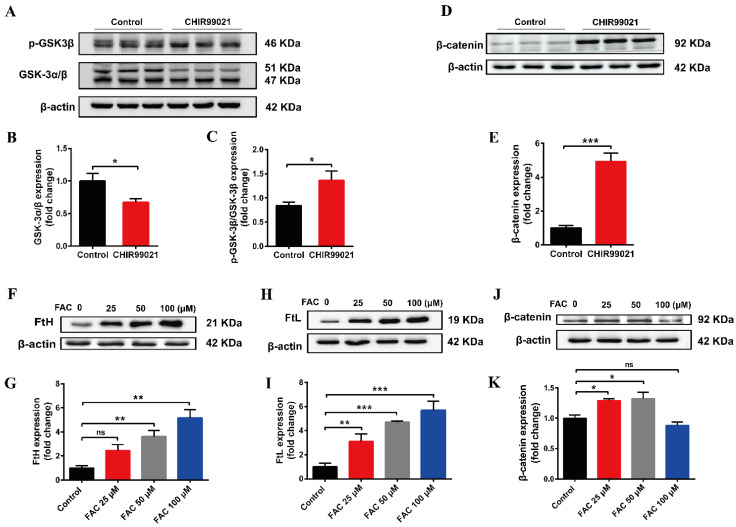
Both CHIR99021 and iron stimulate the expression of β-catenin in Neuro-2a cells. (**A**,**D**) Western blot analysis of GSK-3 α/β, p-GSK3β and β-catenin protein in Neuro-2a cells treated with or without CHIR99021. (**B**,**C**,**E**) Quantification of the Western blot analysis of GSK-3 α/β (**B**), p-GSK3β/GSK-3β (**C**), and β-catenin (**E**). (**F**,**H**,**J**) Western blot analysis of FtH (**F**), FtL (**H**) and β-catenin (**J**) protein after treatment with the indicated concentrations of FAC. (**G**,**I**,**K**) Quantification of the Western blot analysis of FtH (**G**), FtL (**I**) and β-catenin (**K**). The relative expression levels are normalized to β-actin and then expressed as the fold of control group. The data are expressed as the mean ± SEM. * *p* < 0.05, ** *p* < 0.01, *** *p* < 0.001, ns, not significant.

**Figure 6 antioxidants-12-00377-f006:**
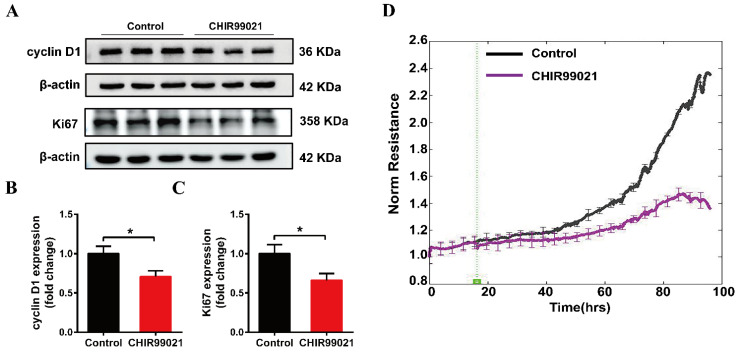
CHIR99021 inhibits the growth of Neuro-2a cells. (**A**–**C**) Western blot analysis and quantification of the expression of cyclin D1 and Ki67 protein. The relative expression levels are normalized to β-actin and then expressed as the fold of control group. (**D**) Detection of cell proliferation by electric cell-substrate impedance sensing. The data are expressed as the mean ± SEM. * *p* < 0.05.

**Figure 7 antioxidants-12-00377-f007:**
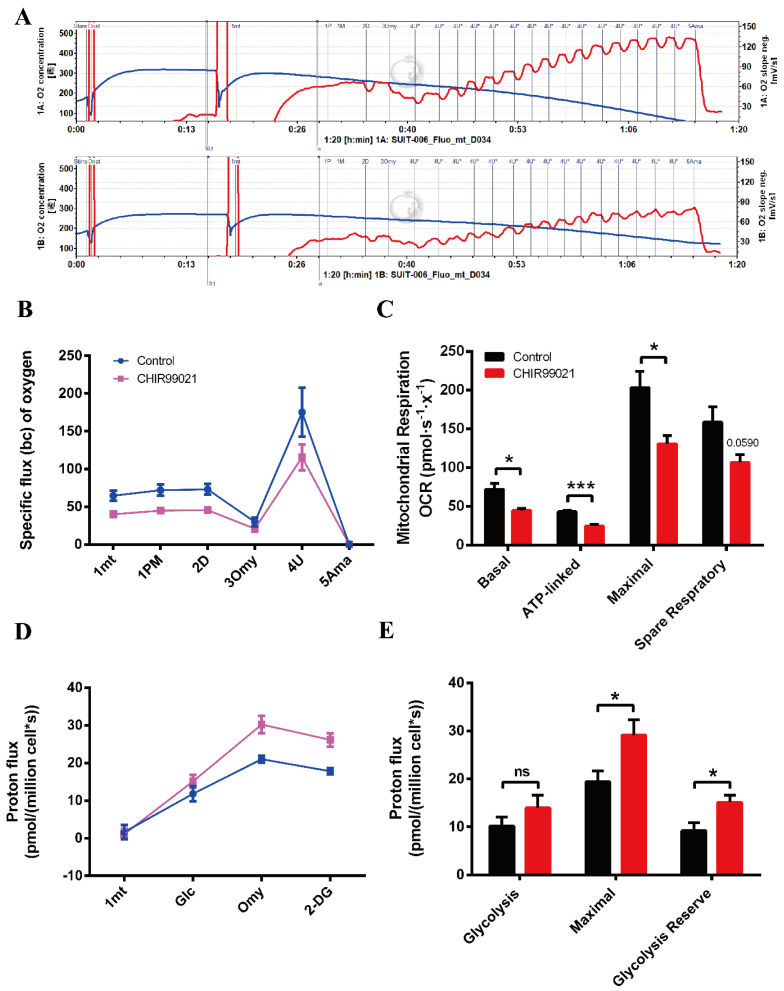
CHIR99021 treatment decreases mitochondrial oxidative phosphorylation and enhances glycolytic capacity in Neuro-2a cells. (**A**) Overall oxygen consumption capacity of cells, as measured by O2k. Top panel, control group; bottom panel, CHIR99021-treated group. (**B**) Oxygen flux of cells after adding the indicated reagents. (**C**) Basal respiration, ATP-linked respiration, maximal respiration and spare respiratory capacity, as indicated by oxygen consumption rate (OCR) in Neuro-2a cells treated as shown. (**D**) Proton flux of cells after adding the indicated reagents. (**E**) Basal glycolysis, maximal glycolytic capacity and glycolysis reserve capacity in cells treated as shown. The data are expressed as the mean ± SEM. * *p* < 0.05, *** *p* < 0.001, ns, not significant.

**Table 1 antioxidants-12-00377-t001:** Primer sequences used for qRT-PCR.

Gene (Gene Accession Number)	Forward (5′-3′)	Reverse (5′-3′)
*TfR1* (NM_001357298.1)	GAGTATCACTTCCTGTCGCCCTATG	GCTGAGAGAGTGTGAGAGCCAGAGC
*FPN1* (NM_016917.2)	TTCCGCACTTTCCGAGATG	AGTCAAAGCCCAGGACTGTCA
*FtH* (NM_010239.2)	TGCCATCAACCGCCAGATCAAC	TCTTCAGAGCCACATCATCT CGGTC
*FtL* (NM_010240.2)	CAACCATCTGACCAACCTCCGCAG	AAAGAGATACTCGCCCAGAGATCC
*Oct4* (NM_013633.3)	GAGGAGTCCCAGGACATGAA	AGATGGTGGTCTGGCTGAAC
*Sox2* (NM_011443.4)	CTGCAGTACAACTCCATGACCAG	GGACTTGACCACAGAGCCCAT
*Steap1* (NM_027399.3)	GGTCGCCATTACCCTCTTGG	GGTATGAGAGACTGTAAACAGCG
*Bola2* (NM_175103.3)	GAACTCAGCGCCGATTACCTC	CAGTGGCTTTCCCTCGAACTT
*Kdm6bos* (NM_001017426.2)	AGTGAGGAAGCCGTATGCTG	AGCCCCATAGTTCCGTTTGTG
*β-actin* (NM_007393.5)	AGGCCCAGAGCAAGAGAGGTA	TCTCCATGTCGTCCCAGTTG

## Data Availability

All of the data is contained within the article.

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
