# Peer review of "CHIR99021 Maintenance of the Cell Stemness by Regulating Cellular Iron Metabolism"

_antioxidants, 2023, doi:10.3390/antiox12020377_

Round 1

Reviewer 1 Report

In the present article titled “CHIR99021 maintenance of the cell stemness by regulating cellular iron metabolism” the authors have shown that CHIR99021 modulates intracellular iron levels through the inhibition of GSK3β while upregulating Wnt/GSK-3β/β-catenin pathway. Manuscript is well written and results are supported by proper experimentation and methodology. Moreover, results were clearly represented and discussed.

Authors need to provide full form of abbreviations at first used place

Microscopy scale and bar is missing.                           

In bar graph, mention non-significant or ns where data is non-significant.

Author Response

Thank you very much for your positive comments and helpful suggestions. We gave a point-by-point reply the queries as the following:

Point 1: Authors need to provide full form of abbreviations at first used place.

Response 1: We thank the reviewer for this valuable suggestion. We have checked the full text carefully and provided abbreviations in full form at first used place (Line 18, line 72, line 249, line 257, line 301, line 307).

Point 2: Microscopy scale and bar is missing.  

Response 2: We are grateful to the reviewer for pointing out this missing. We added the microscopy scale and bar on the Figure 1E and manuscript (Line 251, page 7).

Point 3: In bar graph, mention non-significant or ns where data is non-significant.

Response 3: Thank you very much for your advice. We have checked all of the bar graphs and marked ns in the bar graph where the data is non-significant according to the result description (Please see the Figure 1C、D; Supplementary Figure 1A; Figure 2G、F; Figure 3B、D; Figure 5G、K; Figure 7E).

Reviewer 2 Report

Strengthes: the paper Han et al. provide evidence that a small compound, the aminopyrimidine derivative CHIR99021 interferes with iron metabolism and maintenance of stemness of mouse embryonic stem cells. Removal of CHIR99021 from embryonic stem cell culture medium, or cultivating embryonic stem cells in differentiation medium, in which LIF, PD025901 and CHIR99021 are simultaneously removed, both result in downregulation of genes involved in iron storage (FTNs) and maintenance of cell stemness, which can be counteracted by iron supplementation that otherwise has no effect on the expression of the same genes in stem cells cultured in embryonic stem cell culture medium.

Switching to mouse neuroblastoma Neuro-2a cells, CHIR99021 was used at the same concentration to examine its effect on expression of gene encoding iron transport and storage functions; strikingly, CHIR99021 treatment elevates FTNs- while reducing FPN1- mRNA levels, and CHIR99021 increases intracellular iron quota, suggesting that CHIR99021 could reduce Fe export and increase Fe intracellular storage, which would support cell growth and stemness. Further, the mRNA levels of Steap1, Bola2 and Kdm6bos, all involved in Fe metabolism related functions, increased significantly upon CHIR99021 treatment.

These data are original and interesting. They might be complemented with experiments designed to study cell Fe export (e.g., using hinokitiol, see below) and would benefit from studying DMT1 as Fe cell import gene.

Weaknesses:

Two stemness gene markers were chosen for study; inclusion of nanog, with reference to prior studies using iron chelators (e.g., Ninomiya et al., 2017) could be useful. Alkaline phosphatase staining of mESCs would be more convincing for non accustomed eyes providing some quantification of the data.

“mESCs treated for 72 h in differentiation media (DV)” is what is meant? (line 244) : cells cultured in DV show normal expression of FTNs and stemness markers (?)

The study of CHIR99021 modulation of iron metabolism in Neuro-2a cells could include as well DMT1 (Fe transmembrane importer).

The use a non saturating dose of Fe chelator (DFO) to determine whether CHIR99021 treatment may revert DFO effect on gene expression of iron storage functions (FTNs) produced apparently nonsignificant data, which prevent reaching a firm conclusion. One may wonder whether iron restriction is a suitable experimental setting to examine the influence of Fe cell export. Conversely, the natural product hinokitiol may serve as a surrogate transmembrane iron transporter that restores iron homeostasis in absence of FPN: whether hinokitiol modulates CHIR99021 impact on Fe homeostasis/stem cell development could be worth studying.

The significance of the finding that Fe promotes the expression of β-catenin is unclear: it appears weak and seemingly disconnected from Fe regulation of FTNs. The single dose of CHIR99021 that has been used throughout the study (3 uM) seems very high compared to the IC50 of CHIR99021 for both GSK-3 isoforms (~10 nM) so that it seems possible that other cell activities potentially affected by CHIR99021 may contribute to the results obtained. It seems that the lowest dose of CHIR99021 inducing maximal inhibition of GSK-3 isoforms should be used to test the impact of Wnt classical pathway on iron metabolism. IQ-1 could be used as alternative to CHIR99021 as GSK-3 inhibitor.  Beta-catenin is known to exert a dual role in both maintenance of and exit from the pluripotent state and depending on cell density.

That CHIR99021 inhibits the growth of Neuro-2a cells appears difficult to reconcile with the role of CHIR99021 in support of stemness and glycolytic activity. Source of CHIR99021?

Author Response

Thank you very much for your positive comments and helpful suggestions. we gave a point-by-point reply the queries as the following:

Point 1: Two stemness gene markers were chosen for study; inclusion of nanog, with reference to prior studies using iron chelators (e.g., Ninomiya et al., 2017) could be useful. Alkaline phosphatase staining of mESCs would be more convincing for non accustomed eyes providing some quantification of the data.

Response 1: We thank the reviewer for these insightful comments. According to your suggestion, we detected the expression of nanog in mESCs cells by qPCR. However, we are very disappointed that the expression of nanog gene was very low, and the number of cycles had reached about 34. On the contrary, the number of cycles of cot4 and sox2 were about 21-24. Therefore, we abandoned the study on the effect of 25 µM FAC on the expression of nanog in DV and DV- medium. Previous studies (Navarro et al, 2012) have shown that there are also nanog-negative cell groups in embryonic stem cells, which increased differentiation propensity. In addition, there are a large number of studies supporting the detection of the expression of oct4 and sox2 can be used to evaluate stemness of mESCs (Tapia et al., 2015, Rizzino et al., 2016, Strebinger et al., 2019). Anyway, we are very grateful to the reviewer for their suggestion.

We have quantified the results of alkaline phosphatase staining and added the bar graph of the quantification of the data (Please see the Figure 1F).

Point 2: “mESCs treated for 72 h in differentiation media (DV)” is what is meant? (line 244) : cells cultured in DV show normal expression of FTNs and stemness markers (?)

Response 2:  We are very sorry that we didn't express this clearly. It means that mESCs were cultured in normal embryonic stem cell medium (derivation medium, DV) for 72h. We have changed “differentiation media” to “derivation medium” (Please see line 245, page 7). It contains KnockOutTM DMEM, KnockOutTM serum re-placement, nucleosides, nonessential amino acids, L-glutamine, 2-mercaptoethanol, Pen/Strep, LIF, CHIR99021 and PD0325901. As a control, cells cultured in DV show normal expression of FTNs and stemness markers.

Point 3: The study of CHIR99021 modulation of iron metabolism in Neuro-2a cells could include as well DMT1 (Fe transmembrane importer).

Response 3: We are very grateful to the reviewer for this constructive suggestion. We had done this part of the experiment on the modulation of DMT1 by CHIR99021 in Neuro-2a cells previously. According to your suggestion, we have added this result to the supplementary Figure 2 and described it in the manuscript (Line 265, page 7).

Point 4: The use a non saturating dose of Fe chelator (DFO) to determine whether CHIR99021 treatment may revert DFO effect on gene expression of iron storage functions (FTNs) produced apparently nonsignificant data, which prevent reaching a firm conclusion. One may wonder whether iron restriction is a suitable experimental setting to examine the influence of Fe cell export. Conversely, the natural product hinokitiol may serve as a surrogate transmembrane iron transporter that restores iron homeostasis in absence of FPN: whether hinokitiol modulates CHIR99021 impact on Fe homeostasis/stem cell development could be worth studying.

Response 4: DFO is a classical iron chelator, which has been used in clinical treatment. Considering that 5 μM of DFO has significantly affected the expression of FTNs, we also observed the inhibition of the increase of ferritin expression induced by CHIR99021 with 5 μM of DFO. Although it seems that the difference is not obvious with the naked eyes, the statistics results show that there is a significant change between the control and DFO group (p=0.0438).

We highly appreciate the reviewer's suggestion to use natural product hinokitiol for studying the CHIR99021 impact on Fe homeostasis/stem cell development. According to your suggestion, we would like to carry out research in the future work.  

Point 5: The significance of the finding that Fe promotes the expression of β-catenin is unclear: it appears weak and seemingly disconnected from Fe regulation of FTNs. The single dose of CHIR99021 that has been used throughout the study (3 uM) seems very high compared to the IC50 of CHIR99021 for both GSK-3 isoforms (~10 nM) so that it seems possible that other cell activities potentially affected by CHIR99021 may contribute to the results obtained. It seems that the lowest dose of CHIR99021 inducing maximal inhibition of GSK-3 isoforms should be used to test the impact of Wnt classical pathway on iron metabolism. IQ-1 could be used as alternative to CHIR99021 as GSK-3 inhibitor.  Beta-catenin is known to exert a dual role in both maintenance of and exit from the pluripotent state and depending on cell density.

Response 5: We thank the reviewer for these comments. Yes, as the reviewer mentioned, there is no obvious correlation between iron regulating FTNs and iron regulating β-catenin. In fact, we want to know whether the change of iron level is the key factor for CHIR99021 to regulate β-catenin? To prove this, in this study, we tried to first verify the regulation of iron on FTNs and prove that iron treatment does cause the increase of cell iron level, and then explore the effect of increased intracellular iron on β-catenin. Therefore, we thank the reviewer for this important question. We have changed the ‘ferritin’ of line 342 to ‘iron level’.

For the concentration of CHIR99021, 3 μM is the commonly used concentration in stem cell culture medium (Plesse see reference [29,30]), which has been shown to be important for the maintenance of stem cell stemness. In this study, we want to find out the mechanism of maintaining stem cell stemness at this concentration of CHIR99021. Our data showed that the change of iron level caused by CHIR99021 may be involved in its mechanism of maintaining stem cell stemness. We also found that the study (Plesse see reference [30]) has used CHIR99021 with 3 μM, 5 μM, 10 μM, 15 μM and 20 μM. For the effects of other low concentrations of CHIR99021 on the impact of Wnt classical pathway and on iron metabolism, we thank the reviewer for their suggestions, and we will continue this study in the future.

We are grateful of your suggestion about IQ-1 could be used as alternative to CHIR99021 as GSK-3 inhibitor. IQ-1 has also been used in the culture of mESCs (Miyabayashi et al., 2007). IQ-1 enhances β-catenin/CBP interactions and blocks β-catenin/ P300-driven transcription, keeping mESCs in an undifferentiated pluripotent state. However, in our study, we mainly explore the effect of CHIR99021 on stem cell stemness by affecting iron. In the future, we can make it as a separate topic to study the effect of IQ-1 on iron metabolism in stem cells.

Point 6: That CHIR99021 inhibits the growth of Neuro-2a cells appears difficult to reconcile with the role of CHIR99021 in support of stemness and glycolytic activity. Source of CHIR99021?

Response 6: We are very grateful of your comment. As previous and our studies showed that CHIR99021 play a vital role on the maintenance of stemness. In stem cells, the mode of energy metabolism mainly depends on glycolysis rather than oxidative phosphorylation and most stem cells remain quiescent (Please see reference [37,38]). When stem cells begin to differentiate, mitochondrial metabolism becomes active with increased level of oxidative phosphorylation (OXPHOS) and cell proliferation (Prigione et al., 2015,Mitra et al., 2009,Park et al.,2019). Our results showed that adding CHIR99021 to Neuro-2a cells can reduce the level of oxidative phosphorylation and increase the glycolytic capacity, and glycolysis is the main energy supply mode of stem cells, which indicates that CHIR99021 has an important role in regulating cell energy supply. Moreover, CHIR99021 needs to be added to the normal medium of stem cells, which indirectly indicates that CHIR99021 is essential for maintaining the normal energy metabolism of stem cells. In addition, some studies have shown that most stem cells remain quiescent (reference [37,38]). From our results, the addition of CHIR99021 to Neuro-2a cells inhibited cell proliferation, manifested as a decrease of protein levels of cyclinD1 and Ki67. This indicates an important role of CHIR99021 in regulating cell proliferation. Meanwhile, CHIR99021 is also an essential component in normal stem cell culture medium, which may indicate that CHIR99021 also plays an important role in maintaining the cell cycle of stem cells.

CHIR99021 is an aminopyrimidine derivative and purchased from Selleck (S2924). We have added the source of CHIR99021 in the line of 128.

References:

Navarro P, Festuccia N, Colby D, et al. OCT4/SOX2-independent Nanog autorepression modulates heterogeneous Nanog gene expression in mouse ES cells[J]. The EMBO Journal ,2012, 31:4547-4562.

Tapia N, MacCarthy C, Esch D, et al. Dissecting the role of distinct OCT4-SOX2 heterodimer configurations in pluripotency[J]. Scientific Reports, 2015, 5: 13533.

Rizzino A, Wuebben E L. Sox2/Oct4: A delicately balanced partnership in pluripotent stem cells and embryogenesis[J]. Biochimica et Biophysica Acta (BBA) - Gene Regulatory Mechanisms, 2016, 1859(6), 780-791.

Strebinger D, Deluz C, Friman ET, et al. Endogenous fluctuations of OCT4 and SOX2 bias pluripotent cell fate decisions[J]. Molecular Systems Biology, 2019, 15(9): e9002.

Miyabayashi T, Teo JL, Yamamoto M, McMillan M, Nguyen C, Kahn M. Wnt/beta-catenin/CBP signaling maintains long-term murine embryonic stem cell pluripotency[J]. Proceedings of the National Academy of Sciences, 2007,104(13):5668-5673.

Prigione A, Ruiz-Pérez MV, Bukowiecki R, Adjaye J. Metabolic restructuring and cell fate conversion[J]. Cellular And Molecular Life Sciences, 2015,72(9):1759-77.

Mitra K, Wunder C, Roysam B, Lin G, Lippincott-Schwartz J. A hyperfused mitochondrial state achieved at G1-S regulates cyclin E buildup and entry into S phase[J]. Proceedings of the National Academy of Sciences, 2009,106(29):11960-5.

Park S, Han SH, Kim HG, Jeong J, Choi M, Kim HY, Kim MG, Park JK, Han JE, Cho GJ, Kim MO, Ryoo ZY, Choi SK. Suppression of PRPF4 regulates pluripotency, proliferation, and differentiation in mouse embryonic stem cells[J]. CELL BIOCHEMISTRY AND FUNCTION, 2019,37(8):608-617.